# The associations between adherence to the Mediterranean diet and physical fitness in young, middle-aged, and older adults: A protocol for a systematic review and meta-analysis

**Bruno Bizzozero-Peroni**[1,2,3]*, **Javier Brazo-Sayavera**[3,4], **Vicente Martínez-Vizcaíno**[1,5], **Sergio Núñez de Arenas-Arroyo**[1©], **Maribel Lucerón-Lucas-Torres**[1©], **Valentina Díaz-Goñi**[3,6©], **Isabel Antonia Martínez-Ortega**[1©], **Arthur Eumann Mesas**[1,7]

1 Health and Social Research Center, Universidad de Castilla-La Mancha, Cuenca, Spain, 2 Instituto Superior de Educación Física, Universidad de la República, Rivera, Uruguay, 3 Grupo de Investigación en Análisis del Rendimiento Humano, Universidad de la República, Rivera, Uruguay, 4 Department of Sports and Computer Science, Universidad Pablo de Olavide, Seville, Spain, 5 Facultad de Ciencias de la Salud, Universidad Autónoma de Chile, Talca, Chile, 6 Instituto Superior de Educación Física, Universidad de la República, Maldonado, Uruguay, 7 Postgraduate Program in Public Health, Universidade Estadual de Londrina, Londrina, Paraná, Brazil

© These authors contributed equally to this work.

* Bruno.Bizzozero@uclm.es

## Abstract

### Introduction

A healthy diet and high health-related physical fitness levels may be part of an overall healthy lifestyle. The relationship between adherence to the Mediterranean diet and physical fitness levels has been analyzed in several studies. However, no studies have synthesized evidence on this relationship throughout adulthood. Moreover, in addition to the overall Mediterranean dietary pattern, the associations of individual components of the Mediterranean diet with physical fitness indicators are also unclear.

### Methods

This protocol for a systematic review and meta-analysis was conducted according to the Preferred Reporting Items for Systematic Review and Meta-Analysis for Protocols statement and the Cochrane Collaboration Handbook. Systematic literature searches will be performed in the MEDLINE (PubMed), Scopus, Web of Science, SPORTDiscus and Cochrane CENTRAL databases to identify studies published up to 31 January 2022. The inclusion criteria will comprise observational studies and randomized controlled trials reporting the associations between adherence to the Mediterranean diet and physical fitness levels on general healthy or unhealthy adults (≥18 years). When at least five studies addressing the same outcome are available, meta-analysis will be carried out to estimate the standardized mean difference of physical fitness according to the adherence to Mediterranean diet. Subgroup

**Data Availability Statement:** Our article does not report data and the data availability policy is not applicable.

**Funding:** B.B.-P. was supported by a grant from the Universidad de Castilla-La Mancha co-financed by the European Social Fund (2020-PREDUCLM-16746). M.I.L.-L.-T (2022-PROD-20657) and S.N. d.A.-A. (2020-PREDUCLM-16704) are supported by a grant from the Universidad de Castilla-La Mancha. A.E.M. was supported by a 'Beatriz Galindo' contract (BEAGAL18/00093) by the Spanish Ministry of Education, Culture and Sport. This study was also supported by the Agencia Nacional de Investigación e Innovación (POS_EXT_2020_1_165371). The funders provided support in the form of salaries for authors [B.B.-P., S.N.d.A.-A., M.I.L.-L.-T., and A.E.M.], but did not have any additional role in the study design, data collection and analysis, decision to publish, or preparation of the manuscript. The specific roles of these authors are articulated in the 'author contributions' section. Funder websites: https://www.educacionyfp.gob.es/ https://www.uclm.es/ https://www.anii.org.uy/.

**Competing interests:** The authors have declared that no competing interests exist.

analyses will be performed according to the characteristics of the population, the individual dietary components of the Mediterranean diet and physical fitness parameters as long as there are sufficient studies.

## Ethics and dissemination

This systematic review and meta-analysis protocol is designed for updating evidence on the associations between adherence to overall Mediterranean diet (and specific Mediterranean foods) and physical fitness levels in young, middle-aged, and older adults. Findings from this review may have implications for public health. The results will be disseminated through peer-reviewed publication, conference presentation, and infographics. No ethical approval will be required since only published data will be used.

## PROSPERO registration number

CRD42022308259.

## Introduction

The Mediterranean diet (MD) has been a robust scientific concept in health research for many years [1]. High adherence to the MD has been associated with several beneficial health outcomes, such as reduced overall mortality, reduced risk of some cancers, cardiovascular and neurogenerative diseases and diabetes [2]. Consistent evidence has demonstrated that following an MD is a key factor in preserving favorable health over the entire lifespan [3]. The main characteristics of the MD include an abundance of plant foods (fruits, vegetables, whole-grain cereal, nuts, and legumes), olive oil as the main source of fat, moderate consumption of fish and seafood, and reduced consumption of red and processed meats [4, 5]. The synergistic effect of the dietary components included in the MD scoring systems [4] leads to a favorable nutrient intake (i.e., low contents of saturated and *trans* fatty acids and high contents of unsaturated fatty acids, dietary fiber, vitamins, and minerals) associated with several health benefits such as better metabolic and inflammatory risk parameters [2].

A progressive nutritional transition characterized by a high adherence to the Western dietary pattern (i.e., high consumption of meat products, processed foods, saturated fat, soda, sodium, sugar, and trans-fat) and a decline in adherence to the MD has been observed among adults worldwide, including in Mediterranean countries [6, 7]. The increase in the Western dietary pattern is a growing public health concern because of its relevant contributory factor for obesity, cardiovascular disease, disability, and mortality worldwide [6, 8, 9].

Meanwhile, the average population levels of health-related physical fitness (PF) have been reduced for several years [10], being a strong predictor of deteriorating cardiometabolic health [11] with clear implications and cardiovascular and all-cause mortality [10, 12]. On the other hand, some evidence points to the potential benefits of healthy dietary patterns on PF levels [13, 14], showing the relevance of diet in improving and preserving PF performance, which is an important marker of health status at different time points in adulthood [15, 16].

Although the association between adherence to the MD and PF levels in adults has been analyzed in several studies [14, 17–19], a systematic review with a meta-analytical understanding of how adherence to the MD is associated with PF levels remains unknown. Thus, the aim of this protocol is to provide a detailed plan for conducting a review synthesizing the evidence

regarding the relationship between adherence to the MD and PF levels throughout adulthood (young adults, middle-aged adults, and elderly adults) and to determine which individual dietary components are associated with each PF parameter.

## Methods

### Protocol and registration

This systematic review and meta-analysis protocol was drafted using the Preferred Reporting Items for Systematic reviews and Meta-Analyses for Protocols (PRISMA-P) statement [20] (S1 Table). The systematic review and meta-analysis have been previously registered in PROSPERO (CRD42022308259). It will be conducted according to the PRISMA 2020 guidelines and following the Cochrane Collaboration Handbook [21]. Ethics committee approval and/or informed consent from patients will not be required since no primary data will be collected.

### Eligibility criteria

To be included, studies retrieved from the peer-reviewed literature must report the following: (i) population: healthy or unhealthy adults ($\geq$18 years); (ii) intervention/exposure: the adherence of the MD according to the overall score of different scales (e.g., Mediterranean Diet Score, Mediterranean Diet Adherence Screener) and to specific components (foods and nutrients) of these scoring systems; (iii) outcome: PF components (cardiorespiratory fitness, musculoskeletal fitness, and motor fitness) by using standardized tests; (iv) designs: observational studies (cross-sectional, case–control, prospective/retrospective cohort and longitudinal) and randomized controlled trials; and (v) period: published before January 31$^{st}$, 2022. Moreover, studies will be excluded if they report: (i) duplicate data published in another included study; (ii) diet in terms of intake of single nutrients, food items, and food groups; (iii) special interest group data (e.g., elite athletes or firefighters); (iv) PF measured by self-report; (v) qualitative data; and (vi) data published as conference/meeting abstracts.

### Search methods for study identification

The systematic search will be conducted in MEDLINE (PubMed), Scopus, Web of Science, SPORTDiscus and Cochrane CENTRAL from database inception up to 31 January 2022. No filters will be used in the systematic search. Further studies will be located by additional searches where reference lists of included studies and relevant systematic reviews will be screened for potential relevance. In case of a lack of data, experts will be contacted requesting information.

The electronic database searches will be limited to keywords, title and abstract. The search terms were identified and grouped from the main components (PICO elements) of the research question. To perform the search strategy, free text terms will be used in combination with Boolean operators, as shown in Table 1.

### Data collection and analysis

**Study selection.** All database references will be imported into Mendeley Manager (v1.19.8; Elsevier, London, UK) and checked for duplications. Following this step and based on inclusion/exclusion criteria, two researchers will independently examine the titles and abstracts. The full text of the identified studies will be screened by two researchers independently against the inclusion/exclusion criteria, with consensus required for final inclusion. Discrepancies between researchers will be resolved by reaching consensus or with the

**Table 1. Search strategy for the MEDLINE database.**

| #1 Population | Adult* OR "young adult" OR "middle aged" OR aged OR elderly OR olde* |
|---|---|
| #2 Intervention/ exposure | "Mediterranean index" OR adherence OR "Mediterranean score" OR "Mediterranean diet" OR MedDiet OR "Mediterranean-style diet" OR "Mediterranean eat" OR "Mediterranean food" OR "dietary pattern" OR "diet quality" |
| #3 Outcome | fitness OR "fitness level" OR "physical fitness" OR "physical performance" OR "functional fitness" OR "physical function" OR "muscle strength" OR "muscular power" OR "muscular fitness" OR "muscle endurance" OR "explosive strength" OR flexibility OR "musculoskeletal fitness" balance OR coordination OR agility OR speed OR "motor fitness" OR "aerobic fitness" OR "aerobic capacity" OR "cardiorespiratory fitness" OR "cardiorespiratory endurance" OR "aerobic endurance" |
| **Search strategy:** [(#1) AND (#2) AND (#3)] | |

Proximity operators (*) will be used to search for root words.

intervention of a third reviewer. The results of the search and selection process will be described using the PRISMA 2020 flow diagram (Fig 1).

**Data collection process.** One researcher will perform data extraction on a standardized template, and a second researcher will check for accuracy. If necessary, additional data will be requested from the corresponding authors via email.

The following study-specific data will be extracted: 1) name of the first author and year of publication; 2) country; 3) study design; 4) sample size; 5) participant information (sex and age); 6) adult age group (young, middle-aged, or older adults); 7) MD adherence indices; 8) specific Mediterranean foods; 9) PF component; and 10) main findings. The information will be summarized in a "Table of characteristics" (S2 Table).

**Intervention/Exposure.** The adherence of the MD will be defined with different scoring systems previously identified [4, 22, 23]. Both the earlier (e.g., Mediterranean Diet Score) and newer (e.g., Mediterranean Dietary Serving Score) MD scoring systems, as well as the more widely validated Mediterranean Diet Adherence Screener, will be considered for the assessment of the MD [4].

**Outcomes.** PF refers to the ability to perform daily activities with vigor, as well as the full range of physical qualities that have a relationship with health, such as aerobic capacity or muscle strength [24]. Based on a previous definition [25], PF outcomes for which data will be sought in this review include cardiorespiratory fitness, musculoskeletal fitness, and motor fitness. PF outcomes should be assessed by using standardized tests (e.g., 20-meter shuttle run test, 1.5-mile run/walk test, 12 minutes run/walk test, handgrip strength test, sit-and-reach test) in the general adult population [26].

**Risk of bias in individual studies.** Risk of bias will be independently assessed at the study level by two researchers using the Quality Assessment Tool for Observational Cohort and Cross-Sectional Studies [27] and the Cochrane Collaboration's tool for assessing risk of bias (RoB2) [28]. In case of discrepancies that could not be resolved by discussion, a third reviewer resolved the disagreements.

**Certainty of the evidence.** The Grading of Recommendations Assessment, Development and Evaluation approach will be used for assessing the certainty of evidence and providing recommendations [29]. The GRADE method will be used involving five steps: 1) Assign an a priori ranking of 'high' to randomized controlled trials and 'low' to observational studies; 2) 'Downgrade' or 'upgrade' initial ranking; 3) Assign final grade for the quality of evidence as 'high', 'moderate', 'low', or 'very low' for all critically important outcomes; 4) Consider other factors that impact the strength of recommendation for a course of action; 5) Make a 'strong' or 'weak' recommendation [30].

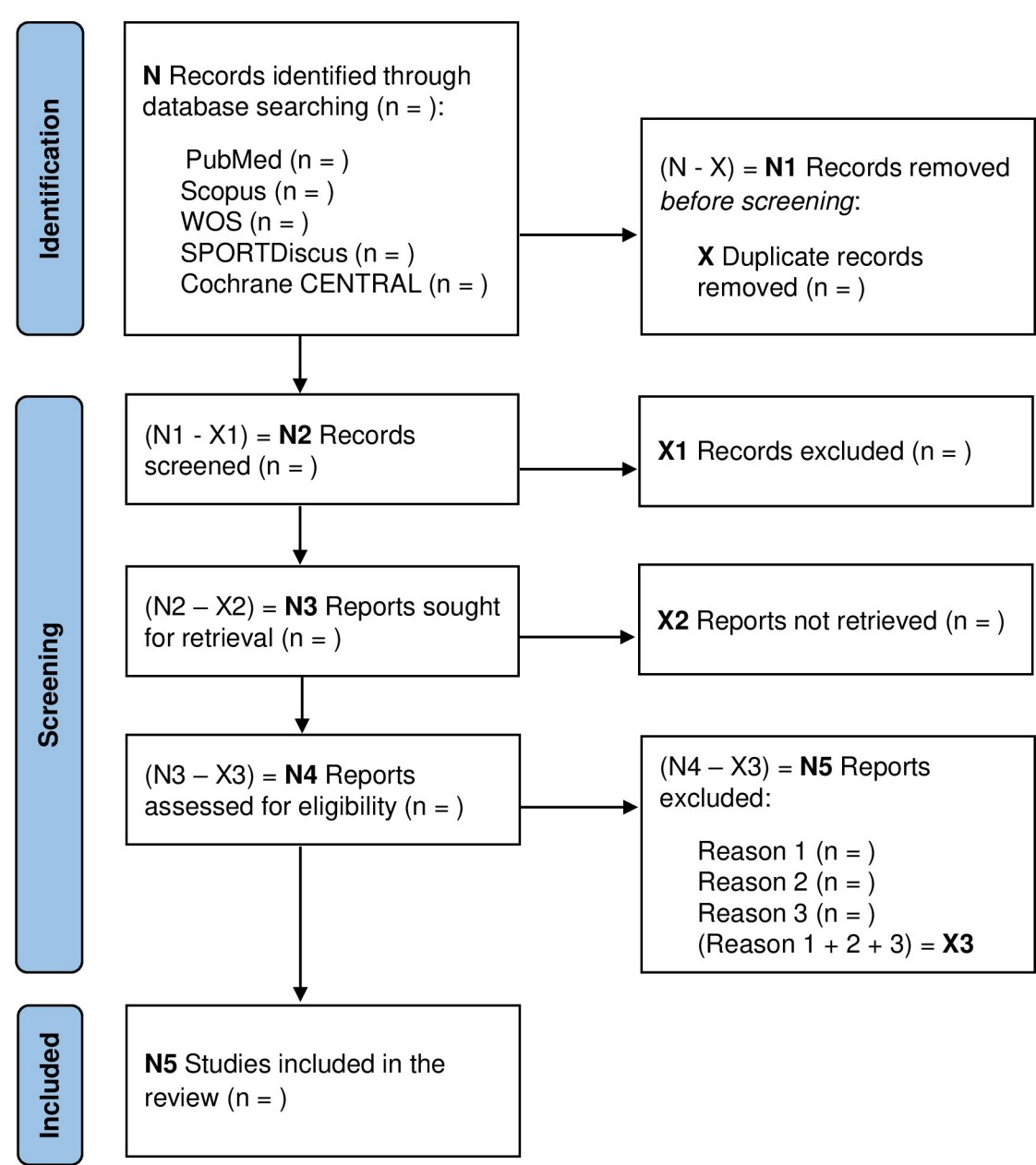

**Fig 1. PRISMA flow diagram for identifying, screening, and determining the eligibility and inclusion of studies.**

### Synthesis of data

Once the main characteristics of the included studies have been extracted, higher vs lower MD exposures will be compared for each PF outcome, and these data will be synthesized narratively sub-grouped by sex, adult age group, and health status. Where a minimum of five studies addressing the same outcome have been identified, a meta-analysis will be conducted. Effect sizes (ESs) and their 95% confidence intervals (95% CIs) will be calculated for each included study using Cohen's d index [31]. If the included studies presented statistical adjustment models, the fully adjusted model will be selected. A pooled ES will be estimated using the DerSirmonian and Laird random effects method [32, 33]. The heterogeneity of results will be

assessed using the $I^2$ statistic, categorizing as not important (0% - 30%), moderate (30% - 60%), substantial (60% - 75%), or considerable (75–100%) (21). In addition, the corresponding $p$ values and 95% CIs for $I^2$ will be considered [34].

If there is available information, subgroup analyses will be performed according to the characteristics of the population (sex, adult age group, health status), intervention (MD scoring systems, individual dietary components of the MD), and outcome (PF tests). Furthermore, the methodological quality of the included studies will be considered for the subgroup analyses.

Random-effects meta-regression analyses will be conducted considering potential main factors causing heterogeneity (e.g., sex, age, study design, body mass index, health status).

To evaluate the robustness of the pooled estimates and detect whether any specific study represents a large proportion of heterogeneity, sensitivity analyses will be conducted by eliminating the included studies one by one.

Finally, publication bias will be tested by visual inspection of funnel plots and using Egger's regression asymmetry test [35].

We will perform all statistical analyses in StataSE software, version 15 (StataCorp, College Station, TX, USA).

## Results

The results of this research will be submitted to a peer-reviewed journal.

## Discussion

This protocol describes the methodology that will be applied for the first systematic review and meta-analysis synthesizing the relationships between adherence to the Mediterranean diet (MD) and health-related physical fitness (PF) levels in young, middle-aged, and older adults. Moreover, the systematic review will intend to provide evidence on the associations of individual components of the MD with PF indicators.

Available evidence indicates that MD is one of the healthiest dietary matrix patterns [36], and PF is an important marker of health in adults [37]. While systematic reviews analyzed the associations between MD patterns and PF levels during adulthood [38–42], they examined only older adults [41, 42], did not specifically analyze adherence to the MD [39, 40], reported results only for one component of PF [40, 42] and did not perform meta-analyses [38–40, 42]. Furthermore, no previous systematic review analyzed the relationships between specific Mediterranean foods and PF levels.

To our knowledge, there are no systematic reviews and meta-analyses that have answered the following questions: Does the available evidence support a relationship between adherence to the MD and PF levels throughout adulthood (young adults, middle-aged adults, and elderly adults)? Which individual dietary components, in conjunction with MD adherence, are associated with each PF component? Since current global dietary transitions have become a growing challenge and public health problem during adulthood, this study may potentially have future implications for public health policies.

The limitations of the review may include the usual limitations of systematic reviews and meta-analyses, such as publication bias, low methodological quality, and heterogeneity of the included studies. Differences among the study designs, sample characteristics, dietary data, PF assessments and methodological quality may restrict comparisons among the included studies and affect the generalizability of the findings.

## Conclusions

This study facilitates the protocol methodology for a systematic review and meta-analysis that will provide updated evidence on the associations between adherence to the MD and PF levels throughout adulthood. Findings from this review may be useful for researchers and health professionals responsible for adult lifestyle surveillance and health promotion. The results obtained will be disseminated through peer-reviewed publications, conferences, symposia, social networks, educational talks, and infographics.

## Supporting information

**S1 Table. PRISMA-P 2015 checklist to address the systematic review protocol, adapted from Table 3 in Moher D et al: Preferred reporting items for systematic review and meta-analysis protocols (PRISMA-P) 2015 statement.** *Systematic Reviews* 2015 4:1.
(PDF)

**S2 Table. Characteristics of studies included in the systematic review and meta-analysis.**
Abbreviations: CRF, cardiorespiratory fitness; MD, Mediterranean Diet; MF, motor fitness; MSF, musculoskeletal fitness.
(PDF)

## Author Contributions

**Conceptualization:** Bruno Bizzozero-Peroni, Javier Brazo-Sayavera, Arthur Eumann Mesas.

**Investigation:** Bruno Bizzozero-Peroni, Sergio Núñez de Arenas-Arroyo, Maribel Lucerón-Lucas-Torres, Valentina Díaz-Goñi, Isabel Antonia Martínez-Ortega.

**Methodology:** Bruno Bizzozero-Peroni, Javier Brazo-Sayavera, Vicente Martínez-Vizcaíno, Arthur Eumann Mesas.

**Supervision:** Javier Brazo-Sayavera, Vicente Martínez-Vizcaíno, Arthur Eumann Mesas.

**Writing – original draft:** Bruno Bizzozero-Peroni, Javier Brazo-Sayavera, Arthur Eumann Mesas.

**Writing – review & editing:** Bruno Bizzozero-Peroni, Javier Brazo-Sayavera, Vicente Martínez-Vizcaíno, Sergio Núñez de Arenas-Arroyo, Maribel Lucerón-Lucas-Torres, Valentina Díaz-Goñi, Isabel Antonia Martínez-Ortega, Arthur Eumann Mesas.

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
