## [Decision Letter · Decision Letter 0]

30 Mar 2022

PONE-D-22-03601The associations between adherence to the Mediterranean Diet and physical fitness in young, middle-aged, and older adults: a protocol for a systematic review and meta-analysisPLOS ONE

Dear Dr. Bizzozero-Peroni,

Thank you for submitting your manuscript to PLOS ONE. After careful consideration, we feel that it has merit but does not fully meet PLOS ONE’s publication criteria as it currently stands. Therefore, we invite you to submit a revised version of the manuscript that addresses the points raised during the review process.

We look forward to receiving your revised manuscript.

Kind regards,

Maria G Grammatikopoulou

Academic Editor

PLOS ONE

Journal Requirements:

"B.B.-P. was supported by a grant from the Universidad de Castilla-La Mancha co-financed by the European Social Fund (2020-PREDUCLM-16746) and from the Agencia Nacional de Investigación e Innovación. 

M.I.L.-L.-T (2022-PREDUCLM-XXX, recent grant-awaiting number contract) and S.N.d.A.-A. (2020-PREDUCLM-16704) are supported by a grant from the Universidad de Castilla-La Mancha.

A.E.M. was supported by a ‘Beatriz Galindo’ contraact (BEAGAL18/00093) by the Spanish Ministry of Education, Culture and Sport. 

The funders had and will not have a role in study design, data collection and analysis, decision to publish, or preparation of the manuscript.

Funder websites:

https://www.educacionyfp.gob.es/

https://www.uclm.es/

" ext-link-type="uri" xlink:type="simple">https://www.anii.org.uy/"

We note that one or more of the authors is affiliated with the funding organization, indicating the funder may have had some role in the design, data collection, analysis or preparation of your manuscript for publication; in other words, the funder played an indirect role through the participation of the co-authors. If the funding organization did not play a role in the study design, data collection and analysis, decision to publish, or preparation of the manuscript and only provided financial support in the form of authors' salaries and/or research materials, please do the following:

a. Review your statements relating to the author contributions, and ensure you have specifically and accurately indicated the role(s) that these authors had in your study. These amendments should be made in the online form.

b. Confirm in your cover letter that you agree with the following statement, and we will change the online submission form on your behalf: 

“The funder provided support in the form of salaries for authors [insert relevant initials], but did not have any additional role in the study design, data collection and analysis, decision to publish, or preparation of the manuscript. The specific roles of these authors are articulated in the ‘author contributions’ section.

Additional Editor Comments:

The reviewers disagree, as one of them accepts the manuscript in current form and the other asks for a revision. We will proceed with a minor revision!

Reviewers' comments:

Reviewer's Responses to Questions

**Comments to the Author**

1. Does the manuscript provide a valid rationale for the proposed study, with clearly identified and justified research questions?

Reviewer #1: Partly

Reviewer #2: Yes

2. Is the protocol technically sound and planned in a manner that will lead to a meaningful outcome and allow testing the stated hypotheses?

Reviewer #1: Partly

Reviewer #2: Yes

3. Is the methodology feasible and described in sufficient detail to allow the work to be replicable?

Reviewer #1: No

Reviewer #2: Yes

4. Have the authors described where all data underlying the findings will be made available when the study is complete?

Reviewer #1: No

Reviewer #2: Yes

5. Is the manuscript presented in an intelligible fashion and written in standard English?

Reviewer #1: No

Reviewer #2: Yes

6. Review Comments to the Author

You may also provide optional suggestions and comments to authors that they might find helpful in planning their study.

Reviewer #1: This work provide a poor rationale for the need of a systematic review including metanalytic pooling of data about the adherence to Mediterranean diet and health-related physical fitness throughout life cycle of adulthood.

However, the manuscript is not suitable for publication as it seems to be a proposal work that was not formatted to meet journal manuscript requirement. The use of future tense in throughout the manuscript specifically in methodology gives evidence of this remark and raise important issue on plagiarism attempt. The methodology section must be fully rewritten and focus on important information related to results and address specifically what was exactly done to obtain the meta-analysis outcomes presented in this work. Most importantly, no results text is available in the submitted manuscript. Moreover, it is stated that “ the results of this research will be submitted to a peer-reviewed journal” .

Abstract shall be written in past tense and shall summarize the research article. I suggest to rewrite the abstract and avoid exhausting methods description but rather providing rational, short description of the methodology, results and impact of the metanalysis review.

Introduction:

Line 82 “ the aim of this protocol is to provide”… not sure if “protocol” suits with the work presented in this research.

Methods:

why authors used future tense? I suggest authors read back the manuscript and be consistent using past tense for descriptive information in this section.

Lines 96- 112Authors shall write a paragraph with sentence and not use bullets to state inclusion and exclusion criteria.

Line 136. Figure 1 cannot be found within the manuscript or not properly labeled.

Line 151. Scoring for adherence to Mediterranean diet shall be briefly described in addition to previous references.

Results: Line 202-203 No results text is available in the submitted manuscript. Moreover, its is stated that “ the results of this research will be submitted to a peer-reviewed journal”

Discussion is too short and with lack of results description, it is not possible to validate arguments raised int his section.

Reviewer #2: This is a very well written protocol, well done!

A few comments:

p.4 line 65, could you please expand?

p.8 line 31: will examine

p. 10 line 177: will be compared

And a suggestion: maybe you would like to present your results per age group. And maybe you could do your analysis per age group? Of course this depends on the evaluation of different outcomes in included studies, but worth thinking about it.

7. PLOS authors have the option to publish the peer review history of their article (what does this mean?). If published, this will include your full peer review and any attached files.

Reviewer #1: No

Reviewer #2: No

---

## [Author Response · Author response to Decision Letter 0]

8 Apr 2022

Maria G Grammatikopoulou 

Academic Editor, 

PLOS ONE 

Enclosed you will find a revision of our manuscript: The associations between adherence to the Mediterranean Diet and physical fitness in young, middle-aged, and older adults: a protocol for a systematic review and meta-analysis. Manuscript Number: PONE-D-22-03601. 

We would like to thank you for giving us the opportunity to revise and improve our manuscript; we also thank the reviewers for their thoughtful and constructive comments. 

We have considered all of the suggestions and have incorporated them into the revised manuscript. Changes to the original manuscript are highlighted in red, and we believe our manuscript is stronger as a result of these modifications. An itemized point-by-point response to the academic editor and reviewers’ comments is presented below. 

Bruno Bizzozero Peroni 

Universidad de Castilla-La Mancha 

Edificio Melchor Cano, Centro de Estudios Socio-Sanitarios 

Santa Teresa Jornet s/n, 16071 Cuenca, Spain. 

E-mail: bruno.bizzozero@uclm.es

Telephone: + (34) 969179100 ext. 4690 

Reviewer(s)' Comments to Editor: 

Editor 

Authors: 

Thank you for the editor comment. We have reviewed and updated our citation style, changing from brackets (5) to square brackets [5]. Moreover, we have resubmitted Figure 1 based on the figures requirements by uploading our figure file to the Preflight Analysis and Conversion Engine (PACE) digital diagnostic tool. 

2. Please review your reference list to ensure that it is complete and correct. 

Authors: 

We have made no changes to the list of references. 

3. We note that one or more of the authors is affiliated with the funding organization, indicating the funder may have had some role in the design, data collection, analysis or preparation of your manuscript for publication; in other words, the funder played an indirect role through the participation of the co-authors. If the funding organization did not play a role in the study design, data collection and analysis, decision to publish, or preparation of the manuscript and only provided financial support in the form of authors' salaries and/or research materials, please do the following: 

a. Review your statements relating to the author contributions and ensure you have specifically and accurately indicated the role(s) that these authors had in your study. These amendments should be made in the online form. 

b. Confirm in your cover letter that you agree with the following statement, and we will change the online submission form on your behalf:  

“The funder provided support in the form of salaries for authors [insert relevant initials] but did not have any additional role in the study design, data collection and analysis, decision to publish, or preparation of the manuscript. The specific roles of these authors are articulated in the ‘author contributions’ section. 

Authors: 

Thank you for the editor comment. We confirmed the follow statement: The funder provided support in the form of salaries for authors [B.B.-P., S.N.d.A.-A., M.I.L.-L.-T., and A.E.M.] but did not have any additional role in the study design, data collection and analysis, decision to publish, or preparation of the manuscript. The specific roles of these authors are articulated in the ‘author contributions’ section. 

Additionally, we have reviewed and updated our statements relating to the author contributions and financial disclosure in the online form. Specifically, we have included the grant number of M.L.-L.-T. 

Authors: 

Thank you for the editor comment. We have revised and changed the Data Availability Statement as follow: Our article does not report data and the data availability policy is not applicable.

---

## [Decision Letter · Decision Letter 1]

28 Jun 2022

The associations between adherence to the Mediterranean Diet and physical fitness in young, middle-aged, and older adults: a protocol for a systematic review and meta-analysis

PONE-D-22-03601R1

Dear Dr. Bizzozero-Peroni,

We’re pleased to inform you that your manuscript has been judged scientifically suitable for publication and will be formally accepted for publication once it meets all outstanding technical requirements.

Kind regards,

Maria G Grammatikopoulou

Academic Editor

PLOS ONE

Reviewers' comments:

Reviewer's Responses to Questions

**Comments to the Author**

1. Does the manuscript provide a valid rationale for the proposed study, with clearly identified and justified research questions?

Reviewer #1: Yes

Reviewer #2: Yes

2. Is the protocol technically sound and planned in a manner that will lead to a meaningful outcome and allow testing the stated hypotheses?

Reviewer #1: Yes

Reviewer #2: Yes

3. Is the methodology feasible and described in sufficient detail to allow the work to be replicable?

Reviewer #1: Yes

Reviewer #2: Yes

4. Have the authors described where all data underlying the findings will be made available when the study is complete?

Reviewer #1: Yes

Reviewer #2: Yes

5. Is the manuscript presented in an intelligible fashion and written in standard English?

Reviewer #1: Yes

Reviewer #2: Yes

6. Review Comments to the Author

You may also provide optional suggestions and comments to authors that they might find helpful in planning their study.

Reviewer #1: The Authors have diligently addressed all the concerns raised by the Reviewers and the manuscript meet PLOSOne requirements to be suitable for publications.

Reviewer #2: There are no further comments from me. The protocol is sufficiently explained and it is ready for publication.

7. PLOS authors have the option to publish the peer review history of their article (what does this mean?). If published, this will include your full peer review and any attached files.

Reviewer #1: **Yes: **Sofi G Julien

Reviewer #2: No

---

## [Editor Report · Acceptance letter]

29 Jun 2022

PONE-D-22-03601R1 

The associations between adherence to the Mediterranean Diet and physical fitness in young, middle-aged, and older adults: a protocol for a systematic review and meta-analysis 

Dear Dr. Bizzozero-Peroni:

I'm pleased to inform you that your manuscript has been deemed suitable for publication in PLOS ONE. Congratulations! Your manuscript is now with our production department. 

Kind regards, 

on behalf of

Dr. Maria G Grammatikopoulou 

Academic Editor

PLOS ONE